# How Advanced Technological Approaches Are Reshaping Sustainable Social Media Crisis Management and Communication: A Systematic Review

**Umar Ali Bukar** [1,2,*,†] , **Fatimah Sidi** [3,*,†] , **Marzanah A. Jabar** [1,†] , **Rozi Nor Haizan Nor** [1,†] , **Salfarina Abdullah** [1,†] , **Iskandar Ishak** [3,†] , **Mustafa Alabadla** [3,†] and **Ali Alkhalifah** [4,†]

[1] Department of Software Engineering and Information System, Faculty of Computer Science and Information Technology, Universiti Putra Malaysia (UPM), Seri Kembangan 43400, Selangor, Malaysia; marzanah@upm.edu.my (M.A.J.); rozinor@upm.edu.my (R.N.H.N.); salfarina@upm.edu.my (S.A.)

[2] Department of Mathematical Science, Faculty of Science, Taraba State University ATC, Jalingo 660213, Nigeria

[3] Department of Computer Science, Faculty of Computer Science and Information Technology, Universiti Putra Malaysia (UPM), Serdang 43400, Selangor, Malaysia; iskandar_i@upm.edu.my (I.I.); gs59711@student.upm.edu.my (M.A.)

[4] Department of Information Technology, College of Computer, Qassim University, Buraidah 51482, Saudi Arabia; a.alkhalifah@qu.sa

\* Correspondence: uabukar@tsuniversity.edu.ng (U.A.B.); fatimah@upm.edu.my (F.S.)

† These authors contributed equally to this work.

**Abstract:** The end goal of technological advancement used in crisis response and recovery is to prevent, reduce or mitigate the impact of a crisis, thereby enhancing sustainable recovery. Advanced technological approaches such as social media, machine learning (ML), social network analysis (SNA), and big data are vital to a sustainable crisis management decisions and communication. This study selects 28 articles via a systematic process that focuses on ML, SNA, and related technological tools to understand how these tools are shaping crisis management and decision making. The analysis shows the significance of these tools in advancing sustainable crisis management to support decision making, information management, communication, collaboration and cooperation, location-based services, community resilience, situational awareness, and social position. Moreover, the findings noted that managing diverse outreach information and communication is increasingly essential. In addition, the study indicates why big data and language, cross-platform support, and dataset lacking are emerging concerns for sustainable crisis management. Finally, the study contributes to how advanced technological solutions effectively affect crisis response, communication, decision making, and overall crisis management.

**Keywords:** crisis informatics; crisis communication; social media; ML; SNA; sustainability

## 1. Introduction

The occurrence of a crisis, particularly a disaster, is hard to predict, but its effects can be minimized through enabling technologies [1]. The advent of crisis or disaster creates uncertainty and chaos among individuals and ruins emergency management recovery plans. A disaster can create an opportunity for fostering sustainable transitions [2]. Likewise, sustainable reporting is intimately connected to crisis response, providing the comprehensive communication necessary to manage stakeholder perceptions [3]. Therefore, accurate and reliable information becomes necessary in such instances to increase community resilience or for sustainable decisions purposes [4]. At the same time, a considerable volume of data about the crisis could be collected through short message service (SMS) from onsite victims, social media, journalists, and humanitarian organizations [5].

Likewise, the period of deep uncertainty created by a crisis such as COVID-19 has encouraged stakeholders to apply best practices and the spirit of open innovation to man-

age the crisis, as reference [6] predicted earlier. Various studies made these predictions practically [7–9]. Specifically, the potential of open innovation approaches in the context of COVID-19 has been investigated considerably [7,8]. Moreover, the work by [9] presented a detailed overview of open innovation in times of crisis. Accordingly, many open innovative activities occur outside of corporate boundaries. The open innovation strategies enable new and confident methods of facing and tackling difficulties [7]. The concept of open innovation, which entails creating and profiting from information through internal and external exchange and transcending organizational barriers [9], is greatly motivated due to a crisis. Therefore, crisis or disaster are the enablers of open innovation, literally. This is because challenges such as the advent of crises create the need to utilize creativity and innovation more than in normal times. As reference [10] rightly put it, there is a need for creativity and innovation to deal with "grand challenges" now more than ever.

For example, as a result of COVID-19, universities are undergoing a radical transformation toward virtuality, revealing a high degree of creativity in terms of continuing academic operations [11]; governments are designing and promoting policies for promoting open innovation [12], numerous innovation projects have been undertaken in the healthcare sector [9], businesses are adopting open innovation strategies to stay competitive [10], small and medium-sized businesses are adopting small subsets of open innovation practices [10], and destination management organizations are adopting open innovation practices [7]. Moreover, crowdsourcing tools (e.g., social networks and hackathons) were identified for practicing effective open innovation in the face of a crisis [8]. Hence, investigating the advances in technological approaches for social media assists crisis management and open innovation strategies in planning effective and sustainable crisis management and decision making.

To avoid confusion, the study first distinguishes between the common approaches used in this study: social media, social network analysis, machine learning, and big data. These approaches greatly impacted the analysis of sustainable crisis communication and management. Firstly, social media applications have been recognized as a reliable communication medium [13], which is defined as an object or environment that enables groups and individuals to collaborate via text, vision, voice, or a combination of these [14–17]. Secondly, social network analysis is a highly effective technique for identifying previously unknown collaboration networks, information flow, and communication between network actors [18]. The word "social network" refers to the structure of social relationships between persons, such as crisis responders [19]. Thirdly, machine learning is a subset of artificial intelligence that enables systems to learn and improve on their own without being explicitly programmed [20]. Fourthly, big data is a term that refers to a large volume and complicated quantity of data that traditional methods are incapable of adequately managing and processing [21–23].

The evolution of social media platforms has demonstrated their effectiveness in assisting crisis-steered impacted communities, and their importance as a source of information cannot be overlooked [24]. A social media insight through content analyses or semantic network analyses can help in applying sustainable crisis response [25,26]. For example, following the 2010 Haiti earthquake, social media analysis became popular, with much of the relief work being conducted through tweets and text messages instead of traditional communication methods [27]. Recent technological advancements have also allowed for applying ML approaches to crisis information management. ML is a powerful tool for crowdsourcing, digital volunteerism, mapping mashups, and recognizing user location [28–30]. Hence, the focus of this study is to evaluate articles that concern crisis informatics to identify how advanced technological approaches are used for crisis responses and recovery to manage the crisis. Thus, this study intends to answer the following questions;

- How is social media related to crisis informatics, and what is the taxonomy of related studies concerning the application of advanced technological approaches in the domain?
- What are the ML techniques used in crisis informatics research and their significance?
- What are the network analysis tools or related techniques used in crisis informatics research and their significance?

Hence, the study is organized as follows: Section 2 discusses the literature review and motivation of the study. Section 3 presents the research methodology, which discusses the paper selection procedure. Section 4 presents the results of the study, which cover the taxonomy and discussion regarding social media and crisis informatics, machine learning tools, network analysis and related techniques, and the focus of these solutions on crisis management. Section 5 discusses the open issues and practices in crisis informatics research. Section 6 discusses the critical reflection and recommendations on crisis management, information management, and decision making. Finally, Section 7 concludes the study.

## 2. Related Review and Motivation

Table 1 summarizes and briefly describes the existing review of crisis informatics research, as well as the focus of related papers based on the categorization of use patterns identified in [31], which examine various practices and tools of social media usage during emergencies. The analysis concentrates on several usage patterns, including digital volunteers, social sensing or analysis of social media, and crowdsourcing.

The development of information technology and new media has resulted in the emergence of social media crisis communication. The majority of existing reviewed articles focus on social media usage and tool-based practice, as well as applications [13,31–42]. A few publications discuss the application of social sensing, crowdsourcing, and digital volunteerism. However, the current studies that examine the application of machine learning or social network analysis are still lacking. Hence, identifying and demonstrating how these emergent techniques shape crisis communication management and improve decision making could assist crisis managers in adopting and implementing effective crisis management and communication strategies.

For instance, Ref. [32] is a pioneering study that gives an overview of early social media crisis and risk communication practices. Ref. [43] examined the sensor web's overlapping domains, including social sensing for public health crises. Ref. [33] undertook a review of the use and impact of social media on crisis management in the tourism industry. On the other hand, Refs. [44,45] are the few review articles that examine social media crisis communication models. Ref. [46] examined social media methods, systems, and applications for disaster management, and [47] examined social media usage patterns, which also presents the data analytics framework with an aspect of big data and social network analysis. Additionally, Ref. [48] examined informal volunteerism, and [34] examines the use of social media in emergency management.

In addition, Ref. [35] focuses on social media usage for information transmission and prediction amid environmental problems, whereas [36] highlights social media usage advancements in data gathering, evaluation, and public participation. Additionally, Ref. [37] classified digital technologies employed in crisis management, whereas [5] examined massive crisis data analytics and supporting technologies such as the internet and mobile phones, crowdsourcing, artificial intelligence, and machine learning. On the other side, Ref. [49] focuses on mobile application engagement during risk and disaster circumstances, while [38] focuses on the exploitation of social media for emergency assistance and preparedness.

Moreover, Ref. [39] investigates the recommendation for effective crisis communication via social media. The authors of [50] assessed the tools and processes used to process catastrophe information. Ref. [40] evaluates and analyses diverse crisis informatics studies, whereas [31] summarized 15 years of social media use in emergencies, focusing on perception, role, and use patterns across various crisis scenarios. Additionally, Ref. [13] focuses on the use of social media in emergency management, Ref. [41] highlights significant ways to use social media for disease outbreaks, and [42] examines the use of social media by low- and middle-income countries in the health sector. Table 1 contains additional information on the topics and themes covered by previously reviewed studies.

**Table 1.** Crisis Informatics Summary of Existing Review Studies.

| Ref. | Focus | Summary | Type |
|------|-------|---------|------|
| [32] | Usage and tools | Social media crisis and risk communication practices | Narrative |
| [43] | Social sensing | The overlapping domains of Sensor Web such as human-in-the-loop sensing and citizen sensing in environmental public health surveillance and crisis informatics | Narrative |
| [33] | Usage | The usage and impact of social media for crisis management in the tourism sector | Narrative |
| [44] | CC models | Social media-based crisis communication theories and how these can be applied to social media crisis communication | Narrative |
| [46] | Tools | Social media methods, applications, and systems for wildfire risk disaster management | Narrative |
| [47] | Tools | Social media use patterns, data analytics framework, data mining tools, associated tools | Narrative |
| [48] | Digital volunteers | The informal volunteerism and the application of social media as volunteerism tool | Narrative |
| [34] | Usage, social sensing, and crowdsourcing | The usage of social media for emergency management, preparedness, technology adoption and usage, information categories, crowdsourcing in disasters, location-based information, as well as disinformation and inaccuracy | Narrative |
| [35] | Usage and tools | Social media usage during environmental concerns for information dissemination, awareness, prediction, promotion, and public participation | Narrative |
| [36] | Usage | Social media usage advances in data collection, public participation, and audience needs evaluations | Review essay |
| [37] | Usage | The usage of digital technologies in crisis management, categorizing the technologies according to usage by crisis management and affected public | Narrative |
| [5] | Usage, tools, social sensing, and crowdsourcing | The use of big data analytics; the processing and analyzing of big crisis data, identified enabling technologies, sources of big data, and its challenges | Narrative |
| [49] | Tools | The applications of diverse mobile apps during risk and disaster situations | Narrative |
| [38] | Usage, tools, digital volunteers, and crowdsourcing | A special issue on "Exploitation of Social Media for Emergency Relief and Preparedness" conducted for the journal of Information Systems Frontiers | Narrative |
| [39] | Usage | Examines the recommendation of researchers for effective social media crisis communication in various strategic communication subdisciplines | Narrative |
| [50] | Tools | The tools and techniques to process the data or information collected after disasters and their challenges | Survey |
| [40] | Usage, tools, social sensing, digital volunteers, and crowdsourcing | Crisis informatics research assessment: types, challenges, trends, and accomplishments in the context of human-computer interaction | Narrative |

**Table 1.** *Cont.*

| Ref. | Focus | Summary | Type |
|---|---|---|---|
| [31] | Usage, tools, social sensing, digital volunteers, and crowd-sourcing | Summarized 15 years of studies about social media usage in emergencies which shows perception, role, and use patterns in different crisis scenarios | Narrative |
| [13] | Usage | The support and complexities of social media application in emergency management | Narrative |
| [41] | Usage | The approaches of using social media in disease outbreaks | Narrative |
| [42] | Usage | Health sector usage of social media in low- and middle-income countries | Narrative |
| [45] | CC models | Social media crisis communication models and their open issues and challenges | Systematic |

Despite a survey of many applications of the crisis informatics literature, to our knowledge, there are only a few studies that focus only on specific themes of machine learning and social network analysis. Hence, we assumed that many tools and techniques, and techniques from machine learning, deep learning, social network analysis, etc., have been introduced to improve crisis management and communication. As a result, this study focuses on reviewing this research gap and its recent advancements in the literature. The research gap was inspired by the lack of a systematic approach to reviewing current crisis informatics studies. Moreover, this study desires to examine the literature about the overlapping integration of ML, SNA, and related advancements in crisis informatics.

### 3. Research Methodology

The review methodology is consistent with the work conducted by [45]. Therefore, prior studies that showed systematic review or systematic mapping approaches are applied with minor differences [51–57]; however, there are overlapping phases or processes, such as the identification of studies, selection, quality assessment, data extraction, and data synthesis [51,52,58–64]. For example, reference [54] splits the selection phase into the development of inclusion or exclusion criteria and selected studies for inclusion and exclusion criteria. Similarly, the articles' inclusion or exclusion criteria are also applied during the title and abstract scanning and full-text reading. Additionally, ref. [56] identified and added reporting as an SLR activity. By considering the various methods for conducting SLR, this study discusses the review methodology in the proceeding sections. The objective of adapting the SLR is to identify the most relevant articles in research and establish a strong background by linking the main aim of this study and prior work.

#### 3.1. Identification of Research and Search Process

Most SLR studies used queries to generate potential articles from various online library databases. The keywords in the search queries help researchers narrow the number of relevant and irrelevant articles and also help in identifying the amount of work conducted in the domain. However, Ref. [29] raised concerns over the lack of disaster-related keywords used to improve effective responses to the people in need. This issue extends to keyword identification for SLR since keywords used in previous studies were not identified. The availability of these could have helped the researchers identify the most relevant keywords for this study. However, Ref. [65] introduced some guidelines to obtain the appropriate keywords adopted and applied in this study. The phrases and keywords include crisis informatics, social media, social media crisis response, emergency management services, response, crisis responses, and crisis communication, as shown in Figure 1. Thus, different queries were developed and applied in six online databases. These databases

include Taylor Francis, Wiley, Springer, ScienceDirect, ACM, and IEEExplore. In addition, a few papers were randomly selected from Google Scholar.

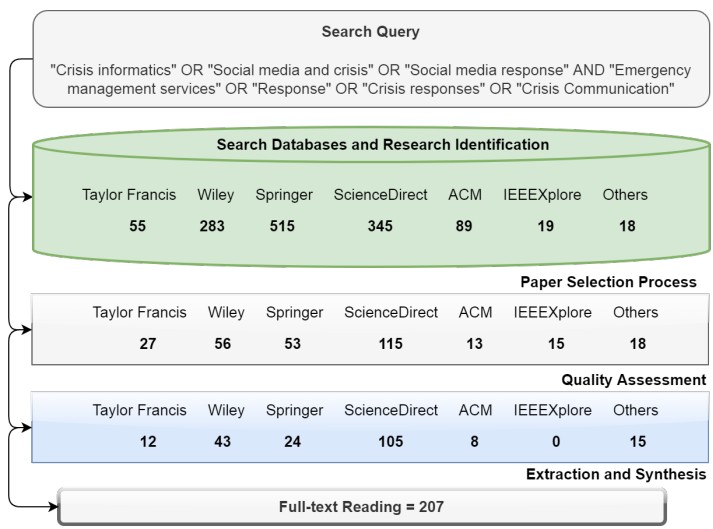

**Figure 1.** Paper selection process.

### 3.2. Paper Selection Process

The selection phase was carried out by title and abstract scanning. The aim is to identify and screen irrelevant articles early before full-text reading. Hence, the inclusion and exclusion criterion was also applied in parallel. The authors identified and adopted exclusion criteria such as non-English text, book chapters, conference papers, and duplicates. Consequently, the inclusion criterion used was indexed IF Journal; any journal not from the journal citation report (JCR) was not considered for further reading. Its rigor influenced the decision to include this criterion for producing high-quality articles. The final sample for full-text reading consists of 207 articles, as shown in Figure 1.

### 3.3. Quality Assessment of Articles

The quality of articles to be considered for review is selected based on predefined guidelines and recommendations [51,57,66]. Reference [51] recommends some guiding principles based on a criterion, while [57] adopted the content validity index (CVI) to assess the quality of papers. The authors computed the CVI value for each article, and the values were used as a benchmark for paper selection. However, this study relied on the quality of JCR publications applied during the paper selection process. Figure 2 presents the frequency of the publications from various journals.

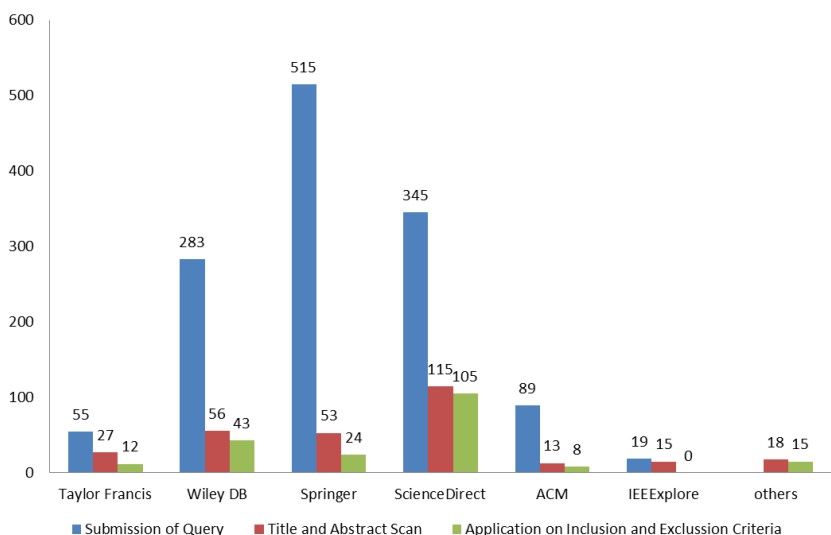

**Figure 2.** Frequency of articles from various journals.

### 3.4. Data Extraction and Synthesis

Data extraction aims to classify and organize the articles based on specific information for excess reading and consideration. The information is categorized based on the recommendation obtained in previous studies [52,66]; this is used in the ranking of published articles. During the organization of the 207 papers, the papers were classified according to "theoretical models", "improvement on theoretical modes", "behavioral models", "frameworks, tools, and systems", "general application and usage", "review papers", and "others". As shown in Figure 3, the theoretical models (green color) were covered extensively by earlier work [45]. Since the study aims to identify the impact of advanced technological approaches in crisis management, articles that proposed frameworks, tools, or systems (red color) were extracted and synthesized. This vital category was not part of the theoretical models of social media crisis communication. Accordingly, while reviewing the frameworks, tools, or systems proposed for crisis management, advanced technological approaches such as ML, SNA, and big data were synthesized. Thus, thematic analysis [54] was adopted and applied to examine and identify commonalities between the selected articles, such as themes, ideas, topics, approaches, or patterns that came up repeatedly.

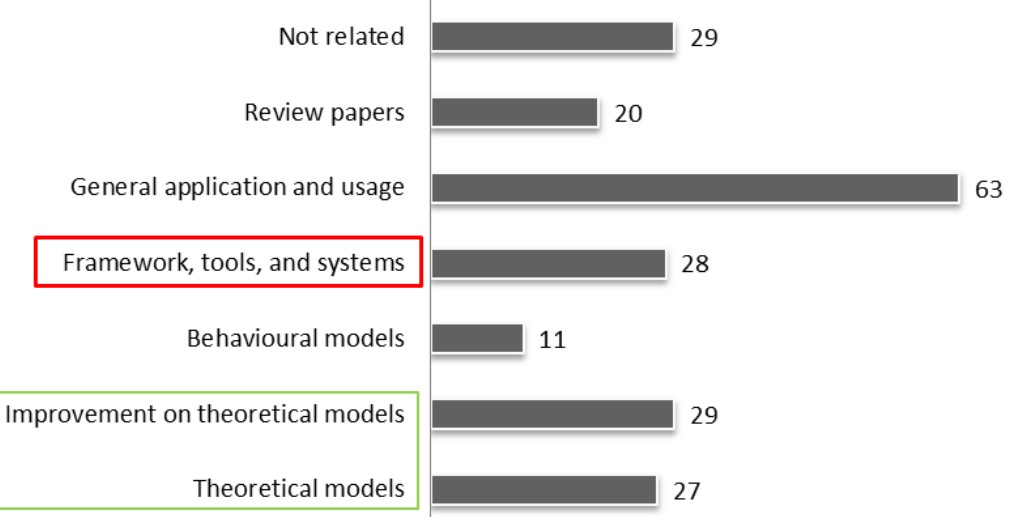

**Figure 3.** Data Extraction and Synthesis.

## 4. Result

As noted, the SLR methodology is consistent with earlier work [45]. Based on the classification presented in Figure 3, this study focuses on analyzing the frameworks, tools, and system classifications against the theoretical model and extension. Hence, a minimum of 28 articles were examined in this study. The result is presented according to the research questions formulated previously.

### 4.1. How Is Social Media Related to Crisis Informatics, and What Is the Taxonomy of Related Studies Concerning the Application of Advanced Technological Approaches in the Domain?

The concept of crisis informatics and its relationship with social media was comprehensively discussed by an earlier study [45]. However, more representation of the taxonomy related to various applications of advanced technological tools is lacking in order to portray how these tools are applicable in crisis management. Therefore, to answer the study's research question, this section presents the current understanding of crisis informatics, then demonstrates the taxonomy of various approaches, particularly ML and SNA techniques identified during the literature evaluation.

Firstly, researchers in crisis informatics identified the term to have been coined by Hagar (2006; 2007) as the concept that uses information technology in crisis management [31,33,49,67–70]. Crisis informatics is a multidisciplinary field characterized by the sociotechnical interactions between people, organizations, and information and technology when a crisis occurs. According to [67], crisis informatics is defined as the empirical study, development, and deployment of information and communications technology (ICT) for crisis management. Ref. [31] stated that crisis informatics refers to social media usage during times of emergency or situational awareness. Accordingly, the field of disaster and crisis studies explores the interplay of social, technical, and information aspects in disasters and crises at all stages, including planning, response, and recovery [68].

Crises are frequently accompanied by an explosion in communication, creating complex information environments. In this type of complex environment, trusted information becomes critical now that crisis informatics bridges the divide between computing and social science knowledge about disasters [70]. This enables the use of ICT to deal with uncertainty in the aftermath of a disaster. In addition, this allows the analysis of online participation to be aided by computational tools. Numerous technologies have contributed to crisis informatics maturity becoming what it is today. Emergency management is viewed through the lens of crisis informatics as encouraging information dissemination within and between crisis managers, public channels, and entities as a sociotechnical system through which information is communicated [69]. Thanks to advancements in computing technology, especially social media, the broad usage and study of crisis informatics have been made possible.

The importance of ICT in crisis management cannot be overemphasized [71]. The phases of crisis, such as prevention, preparedness, response, and recovery, can be used to categorize systems and software [72]. In the context of disaster response, social media builds open crisis; informatics views emergency response as an expanded social system. This encouraged stakeholders (e.g., the general public and emergency managers) to generate and disseminate disaster-related information to a broader audience [73]. In times of crisis, citizens commonly utilize social media to communicate with one another and share information about the situation. At the same time, emergency operation centers frequently use it as a source of information to increase their situational awareness [74]. Social media has become so interwoven in our everyday lives, and it is essential to underline the relevance of social media as a resource [15]. People are increasingly turning to the internet for information during times of distress [75]. With the support of external actors, improved crisis communication is vital to crisis management success [76]. During crises, the digital convergence of people, information, and resources have been thoroughly documented in multiple articles in the crisis informatics literature [67,77].

Furthermore, according to [78], data mashups are a type of social media that allow data generation through an application programming interface (API), allowing users to combine functionality or feeds from other websites. Because of social media, the general population no longer acts as a passive recipient of information; instead, they actively seek out crisis information and share their thoughts with others [79]. Friends and relatives are excellent sources of information about a crisis [80]. When the person giving the warning message is in a similar social position to their own, individuals are more likely to take the information seriously [81]. The participation of general people in disaster/crisis response through social media is not a new concept. Social media amplify the public's response [82]. Effective crisis communication and management necessitate both formal and informal stakeholders (management and the general public) [82,83].

The widespread adoption of technology has enabled crisis response and humanitarian development to be considered the future of human progress and well-being [5]. The advances in computing, communications, storage, processing, and analysis have driven crisis management. Technology-driven emergency management is a new research topic that is constantly expanding. Each step forward in improving procedures or tools can substantially contribute to saving human lives and resources. Frequently, the terms "emergency management", "disaster management", and "crisis management" are used interchangeably [84]. In contrast, sustainability communication is a relatively new field with sometimes ambiguous limits [85]. Nevertheless, the role of disaster/ crisis management is to coordinate efficient actions related to multiple missions such as information, security, supply, and lodging in a highly dynamic and uncertain environment [86].

Figure 4 summarizes the taxonomy of technology solutions techniques identified from the current literature. Interestingly, most of these tools are used in open innovation strategies. Accordingly, the advanced technological solutions for crisis management are classified into crowdsourcing tools, social media, big data, ML, news and blog platforms, SNA, and mashup tools, as shown in Figure 4. Moreover, the ML is further classified into applications and classification tools. In contrast, social network analysis is divided into mining tools (API, code frame, and constitutive rule) and text analysis tools (semantic analysis, thematic analysis, and textual analysis), as well as techniques such as Getis-Ord Gi* statistics and software for visualization. Network analysis is used to understand the network structures of the actors (the management and public) involved in crisis communication. Public actors are classified into influencers, followers, and inactives, known as first public, second public, and third public [14].

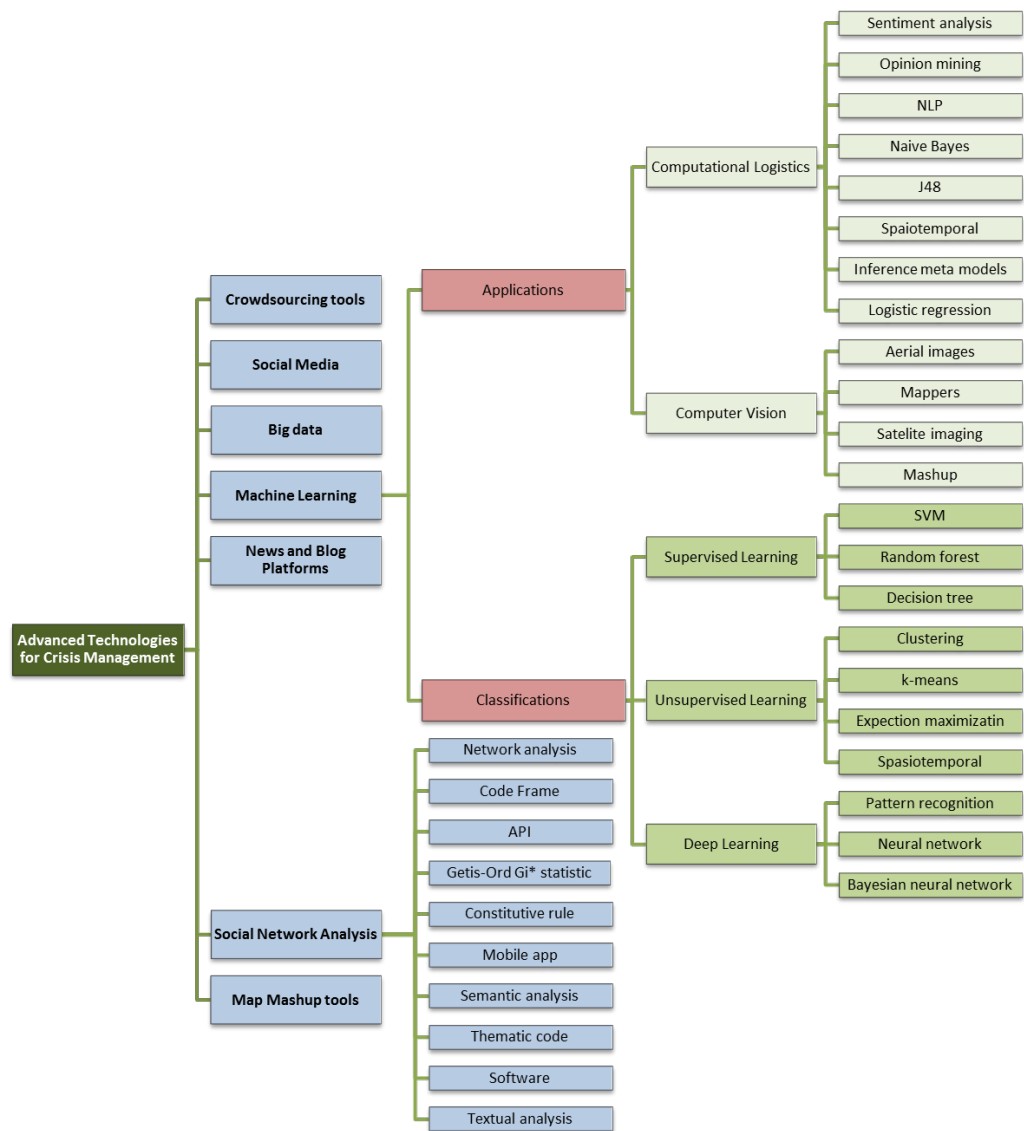

**Figure 4.** Taxonomy of advanced technological approaches in crisis informatics.

Accordingly, several studies integrated machine learning and social network analysis to predict health-related emergencies, predict patient experience [19], and determine the evolution and sentiment polarity of albinism [87]. Moreover, ML and SNA were used to characterize topics and communication network dynamics movement during a crisis [88], as well as were applied to reveal organizational structures [89]. The literature shows the significance of these tools and why researchers in crisis management and communication should have an instantaneous grasp of how these tools are shaping the research area.

There is considerable confidence that big data techniques will provide insight into the rapidly changing circumstances and aid in the development of efficient disaster response [5]. SNA can potentially improve the performance of machine learning algorithms [89]. SNA is a quantitative and qualitative method for extracting data and analyzing the structural properties of networks [90]. In this regard, the advent of social media and the subsequent growth in social media users has resulted in an explosion of user-generated content (UGCs). Social media are key big data providers and match the definition of big data [91]. Thus, it is critical to understand what big data is and how it may be used to obtain insight [92].

### 4.2. What Are the ML Techniques Used in Crisis Informatics Research and Their Significance?

The articles that focus on the ML concept were screened and examined to address this research question. Specifically, the ML techniques were identified (Table 2). Furthermore, Table 3 summarizes the crisis management problem addressed by the existing systems, as well as their corresponding data sources.

#### 4.2.1. Application of Machine Learning Tools or Techniques

The evaluation and synthesis of the literature furnished this study with the various application of ML techniques to improve crisis management effectiveness. Ref. [93]'s structural framework was the foremost algorithm that visualized and filtered crisis-related tweets. The self-organizing map algorithm was implemented using artificial intelligence features for decision support. The framework adopted Twitter to study the reactions, communications, and sentiments of the public's response to terrorism. The information extracted from the framework can be used by crisis management to improve rapid response, detection, and recovery.

**Table 2.** ML Tools and Techniques in Crisis Informatics Research.

| No | Ref. | Name of the System | ML Tool or Techniques | Description |
|---|---|---|---|---|
| 1 | [93] | Structure Framework | Self-organizing map algorithm | Visualization and filtering |
| 2 | [94] | GeoCOVANI Apps | J48 and naive Bayes | Retrieve, process, and analyze social media content |
| 3 | [4] | Named entity recognition software + ML techniques | NLP, Lexio-semantic pattern, and matching | Location identity algorithm |
| 4 | [95] | Decision tree classification model | ML (Decision tree classifier), chi-squared test, and ten-fold cross-validation | Classification of tweet conversation |
| 5 | [82] | Twitris system | NLP, semantic analysis, lexical and syntactic analysis, and spatiotemporal thematic visualization | Identify information seekers and suppliers |
| 6 | [96] | Analysis system | Sentiment analysis, time-series analysis | Expression of fear and social support |
| 7 | [97] | Visual analytics system | Spatiotemporal analysis | Public behavior analysis and response planning |
| 8 | [77] | Crisis crowdsensing Framework | Spatiotemporal | Key dimensions of crisis crowdsensing |

**Table 2.** *Cont.*

| No | Ref. | Name of the System | ML Tool or Techniques | Description |
|---|---|---|---|---|
| 9 | [98] | Baseline assessment | Spatiotemporal pattern | Analyzing content |
| 10 | [99] | Anomaly Detection Approach | Spatiotemporal filtering | Detect rumoring during disaster events |
| 11 | [24] | Hybrid human–agent systems | Modular Bayesian network + Inference metamodels | Situational assessment framework |
| 12 | [28] | Algorithm | SVM, Social network analysis | Computed regression-based time series |
| 13 | [100] | Urban safety app | | Sensor node |
| 14 | [29] | Real-time text classification algorithm | SVM and linguistic features | Framework based on a hybrid method for classification |
| 15 | [30] | Sentiment classification and categorization model | SVM and sentiment analysis | Information management |
| 16 | [101] | Exodus 2.0: Crowdsourced data | Spatiotemporal and thematic | Analysis of migrants pathways |
| 17 | [102] | Spatial Mercalli scale prediction system | Support vector regression | Early prediction system based on social media features |
| 18 | [84] | Recommendation and DSS | Rule based | Risk and crisis assessment |
| 19 | [103] | Location reference identification using deep learning | Convolutional neural network (CNN) | Identification of tweet location |
| 20 | [104] | National-scale Twitter data mining pipeline | NLP, machine learning libraries, logistic regression, and spatiotemporal | Identify the location of tweets |

Moreover, reference [94] developed the GeoCOVANI app based on J48 and naive Bayes machine learning techniques on the Weka software suite that supports rule set [29,94]. The GeoCOVANI system retrieves, processes, and analyzes social media content to support disaster information management. Ref. [4] adopted a "named entry recognition software" system embedded with ML techniques. The software works with natural language processing (NLP) to identify references to street addresses, buildings and spaces, toponyms, place acronyms and abbreviations, and ML for abbreviation disambiguation to find buildings. The motivation behind named entity recognition software is that the location of an author's social media message is not the same as the location the author writes about in the message. However, in a disaster/crisis, content geolocation is more important than the author's location. The reference [4] effectively geo-parses microtext (the short and informal messages), and their investigation shows that many microtext messages contain abbreviated, misspelled, or highly localized references. The algorithms identify the actual location of the content of text posted on Twitter rather than the user location.

Furthermore, the literature shows the use of decision tree ML techniques for crisis management solutions [29]. Moreover, the work by [82,95] was an early study that proposed a decision tree classification model based on an ML decision tree classifier, chi-squared test, and ten-fold cross-validation. The model was used to support the classification of tweet conversation for detecting and analyzing citizen cooperation during the recovery phase of a crisis. However, this model was limited to English and could require extensive revision to extend it to other languages. Therefore, Ref. [82] developed a "Twitris" system based on NLP, semantic analysis, lexical and syntactic analysis, spatiotemporal, and thematic

visualization. Twitris aims to identify information seekers and suppliers during disaster/crisis to support crisis coordination. The result shows that their approach met informal and formal crisis response expectations for effective collaboration. The system is expected to coordinate the emergency response for resource distribution and enhanced situational awareness [82].

Moreover, Ref. [96] developed a system for sentiment analysis and time-series analysis to determine the expression of fear and social support from tweets. The motivation of the study is related to the analysis and identification of intercommunity support for those affected by disasters. The study analyzes the reaction of Twitter users on how they expressed three types of emotions (fear, sympathy, and solidarity) regarding terrorism [96]. The literature evaluation also identified some studies that use spatiotemporal techniques inclusive of Exodus 2.0 [101] and the national-scale Twitter data mining pipeline [104]. In addition, Ref. [97] used visual analytics system-based spatiotemporal analysis to study public behavior in order to implement effective response planning and decision making using social media data. The system demonstrated an analytical scheme from multiple spatial data sources and enabled the verification and evaluations of abnormal situations. Moreover, Ref. [77]'s crisis crowdsensing framework uses the spatiotemporal concept in an emergency management system. The framework was validated using data from qualitative and interview research strategies. Reference [98] proposed a baseline assessment based on the spatiotemporal pattern to analyze early warning content and disaster planning. In addition, Ref. [99]'s anomaly detection approach is also based on spatiotemporal filtering. The main objective of [99] is to detect rumoring during disaster events for crisis communication and decision support. Hence, the work by [86] was not the only system that used humans and artificial agents. Modular Bayesian networks and inference metamodels were proposed as hybrid human–agent systems [24]. The hybrid system is a situation-based assessment framework for crisis management and decision support that combines humans and artificial agents as sensors for crisis management.

Additionally, the work by [100] developed an app with artificial intelligence capability, called "Urban safety", based on intelligent sensor nodes and crowdsensing data collection. Although the artificial intelligence (AI)'s urban safety technique is not straightforward, the AI feature suggests that the system falls within this category. The studies by [29,30] proposed a real-time text classification algorithm [29] and sentiment classification and categorization model [30] for situational awareness, information management, and decision support. The real-time text classification algorithm combines a support vector machine (SVM) and linguistic features. The proposed hybrid method combines rule-based methodology and machine learning algorithms with linguistic features for segregating the texts and classifying them according to needs [29]. Similarly, the sentiment classification and categorization model employed SVM and sentiment analysis to collect disaster data and categorize them based on affected people's needs. The ML algorithms use the categorized data and classify them to analyze people's sentiments [30].

Understandably, the SVM is a popular machine learning technique that uses computational logistics. Moreover, the SVM is a supervised machine learning algorithm used for classification and regression analysis [29]. Remarkably, the [28] algorithm is based on the SVM, social network analysis, and computed regression-based time series. The tweet behavior was analyzed and revealed by regression-based time series. At the same time, the ML method identifies uncertainty in tweets, and the network analysis was employed to determine the most influential actors and their respective social positions. The result of the study shows how social position affects collective sense making in crises.

Notably, Exodus 2.0 by [101] is based on spatiotemporal and thematic analysis. The Exodus system supports the analysis of migrants' pathways. Accordingly, Exodus 2.0 was coined to refer to the digital age of the new migration paradigm. In addition, Ref. [102] proposed a spatial Mercalli scale prediction system and adopted the SVM regression with a sequential minimal optimization (SMO) algorithm on Weka. The spatial Mercalli system supports early prediction based on people's reactions to earthquakes on social media data

to support emergency response. The scale prediction system was trained using five-fold cross-validation, applied instance resampling to deal with intensity-unbalanced data, and utilized normalized polynomial kernel to calculate the support vectors.

**Table 3.** Problems Addressed by the Existing System, Method, and Data Source.

| No | Ref. | Problems Addressed by the System | Method | System Developed by the Author | Data Source |
|---|---|---|---|---|---|
| 1 | [93] | Decision making | Experiment, simulation | 1 | Twitter |
| 2 | [94] | Disaster information management | Prototype + Case study | 1 | Twitter, Flickr, EFFIS |
| 3 | [4] | Decision supports | Experiments + case study | 2 | Twitter |
| 4 | [95] | Citizen cooperation detection | Experiment | 1 | Twitter |
| 5 | [82] | Support crisis coordination | Experiments + case study | 1 | Twitter |
| 6 | [96] | Intercommunal emotions and expressions | Experiment + Case study | 0 | Twitter |
| 7 | [97] | Decision support | Case study | 1 | Twitter |
| 8 | [77] | Emergency management system | Qualitative + interview | 1 | Twitter |
| 9 | [98] | Early warning and planning for disaster | Unclear | 0 | Twitter |
| 10 | [99] | Crisis communication and decision support | Case Study | 1 | Twitter |
| 11 | [24] | Environmental crisis management and decision support | Prototype | 1 | Twitter |
| 12 | [28] | Social position and collective sense making | Case study | 1 | Twitter |
| 13 | [100] | Crowdsensing: urban data collection | Experiment | 1 | Unclear |
| 14 | [29] | Situational awareness | Experiment + case study | 1 | Twitter |
| 15 | [30] | Decision support | Experiment | 3 | Twitter |
| 16 | [101] | Data and information management | Case study | 0 | Twitter, Flicker |
| 17 | [102] | Emergency response | Experiment + case study | 1 | Twitter |
| 18 | [84] | Critical infrastructure | Case Study | 1 | App message |
| 19 | [103] | Location-based services | Experiment | 1 | Twitter |
| 20 | [104] | Situational awareness | Prototype + case study | 1 | Twitter |

Furthermore, a decision support system was proposed for risk and crisis assessment of critical infrastructure [84]. However, reference [103]'s study is one of the earlier studies that adopted deep learning approaches in crisis management and communication. The study conducted location reference identification based on a convolutional neural network (CNN), which identifies the location of tweets for crisis management purposes. The deep learning-based model used Twitter API to collect earthquake information to help early event localization, emergencies, real-time road traffic management, localized advertisement, and various location-based services. NLP was seen to widely support ML advancement. A national-scale Twitter data mining pipeline that combines NLP, machine learning libraries, logistic regression, and spatiotemporal techniques was introduced to identify the location of tweets to support situational awareness efforts [104]. The prototype-based pipeline is a modular architecture that integrates web data sources to obtain at-risk locations that enable real-time retrieval of geotagged tweets. In addition, the NLP and machine learning libraries allow filtering of flood-relevant tweets.

### 4.2.2. Summary

The analysis of existing studies showed the effectiveness of machine learning tools and techniques in crisis communication and management. Different solutions grounded

in ML were used to improve various areas in the crisis management domain. For example, ML improves the visualization and filtering of crisis-related posts. Moreover, ML is used to retrieve, process, and analyze social media content for knowledge purposes. Furthermore, ML was used for location identity, classification of crisis response posts, identifying information seekers and suppliers, detecting rumoring during crisis events, and finding out the key dimensions of crisis crowdsensing. Moreover, understanding public behavior help improves crisis response planning. Nevertheless, the analysis showed how the ML concept identifies or classifies users' posts that express fear in order to enable crisis managers to provide social support to victims. In addition, the ML application in crisis management helped improve the effectiveness of situational assessment, information management, early prediction, and risk and crisis assessment. Hence, addressing these issues through ML collectively helped plan, communicate, manage, and coordinate effective and sustainable crisis management and decision-making processes.

*4.3. What Are the Network Analysis Tools or Related Techniques Used in Crisis Informatics Research and Their Significance?*

The articles that focus on SNA or other techniques were screened and examined to address this research question. Specifically, SNA and related techniques are identified in Table 4. Furthermore, Table 5 summarizes the crisis management problem addresses the existing systems, as well as their corresponding data sources.

4.3.1. Application of Network Analysis and Related Techniques

The study also identified some helpful studies which are not based on ML techniques. Interestingly, some of these studies focus on using SNA or systems for crisis management. The systems and framework used in disaster response and recovery have three main phases: data collection, processing, and output [29]. Various tools and techniques are used in these phases. For example, ref. [105] recently proposed a framework based on situational awareness based on social media data for decision making [106]. Table 4 presents the summary of network analysis applications in crisis informatics research.

**Table 4.** Application of Network Analysis and Related Techniques.

| No | Ref. | Name of the System | Techniques | Description |
|----|------|--------------------|------------|-------------|
| 1 | [73] | Science gateway platform | Network analysis | Web application and Spatial DSS |
| 2 | [107] | Virtual research environment (VRE) | Code frame | Analyze Twitter data |
| 3 | [71] | Interactive spreadsheet | | Collaboration and disaster info sharing |
| 4 | [105] | Cross-source triangulation framework | Flickr and Twitter API, Getis-Ord Gi* statistic | Impact area assessment |
| 5 | [86] | Situated artificial institutions | Constitutive rule | Hybrid, mixed, and normative multiagent approach |
| 6 | [108] | Human as a sensor (Haas) | Social network analysis | Detect unfolding earthquakes |
| 7 | [109] | iCEMAS | | Knowledge management and awareness system |
| 8 | [76] | City-share apps | Mobile app | Public display application |
| 9 | [110] | Index dictionary | Semantic analysis | Identification of damage-related tweet |
| 10 | [111] | Internet memes | Network analysis | Map user's activity and types of user identifications |
| 11 | [112] | Humans of New York (HONY) blog | Thematic code | Observational ethnographic methods |
| 12 | [74] | Semantic visualization tool | Software | Visual analytics |
| 13 | [113] | Console framework | Textual analysis, semantic | Communicating bad news |

The study by [73] provided a web application platform, 'Science Gateway', which conducted network analysis using spatial data for decision-making purposes. The virtual

research environment (VRE) was proposed by [107] based on a relational database that analyses Twitter data for decision support. The VRE was implemented to support information flow and rapid access to the database. Furthermore, ref. [71] proposed a platform based on an interactive spreadsheet. The spreadsheet aims to support collaboration and sharing of disaster information for emergency response and decision making. Information exchange between onsite responders and central decision makers is improved using the spreadsheet-based collaborative system, enhancing situational awareness. Mobile devices enable users to collaborate and exchange portions of the data, apply for resources concurrently and reliably, and receive real-time status updates from decision makers through the system.

Remarkably, the work by [105] adopted a cross-source triangulation framework to assess the statistical impact of natural disasters. The triangulation framework's main objective is to integrate social media disaster content to geolocate and track the disasters and content contributors. A situated artificial institution based on the constitutive rule with a normative multiagent approach was employed by [86] to improve the regulation of human collaboration during the recovery phase of a disaster. Accordingly, the study proposed a multiagent method to cope with dynamic, distributed, and decentralized crisis management. Human and artificial actors interact in this approach to regulate mixed interactions [86]. Similarly, Ref. [108] used human as a sensor (Haas) to analyze and detect unfolding earthquakes for a decision support system. The Haas conceptual and architectural framework was demonstrated via Twitter and validated experimentally on messages posted during earthquakes.

Furthermore, a prototype called iCEMAS was introduced and used for information management, dissemination, and awareness [109]. Moreover, an app called "City-Share" was developed for community disaster resilience and was tested through a prototype and an empirical method [76]. The result indicates that the app is effective for awareness messages. The app (City-Share) acts as a sensor that collects urban data, such as structural acceleration, structural deformation, questionnaires, and images. Furthermore, it implements disaster emergency communications without the use of a network. Hence, the summary of articles based on SNA and related techniques is presented in Table 4.

**Table 5.** Problems Addressed by the Existing System, Method, and Data Source.

| No | Ref. | Problems Addressed by the System | Method | System Developed by the Author | Data Source |
|----|------|----------------------------------|--------|-------------------------------|-------------|
| 1 | [73] | Structural information dissemination | Unclear | 1 | App message |
| 2 | [107] | Decision support | Experiments | 1 | Twitter |
| 3 | [71] | Emergency response and decision making | Artefact + Experiment | 1 | App message (Teams data) |
| 4 | [105] | Management of natural disaster | Case study | 1 | Twitter, Flickr |
| 5 | [86] | Regulation of human collaboration | Unclear | 1 | |
| 6 | [108] | Decision support system | Experiments | 1 | Twitter |
| 7 | [109] | Data management, information sharing, and dissemination | Prototype, model | 2 | Twitter, Facebook, |
| 8 | [76] | Community disaster resilience | Prototype + Interview | 1 | App message |
| 9 | [110] | Crowdsourcing: damage assessment | Case study | 0 | |
| 10 | [111] | Stakeholders' resistance to terrorism | Unclear | 0 | Twitter |
| 11 | [112] | Marketing and branding | Case study | 0 | Facebook |
| 12 | [74] | Filter and visualize relevant information | Experiments | 1 | Twitter |
| 13 | [113] | Crisis communication | Case study | 0 | Twitter, Facebook |

Additionally, ref. [110] proposed an indexed dictionary based on semantic analysis to identify damaged tweets from stakeholders. The index dictionary aimed to assess damage levels. Furthermore, ref. [111] adopted a network analysis method to determine how stakeholders used internet memes to resist terrorism, and users posted kittens instead of lockdowns. The network analysis maps users' activities and identifies them based

on their role, identifying user profiles and active profiles in producing memes. Finally, ref. [74] developed software for semantic visual analysis to visualize relevant information for crisis management.

Moreover, Ref. [113] adopted the Console framework based on textual analysis for communicating bad news. The result shows that the Console framework is practical for crisis communication and helps organizations better understand how to share bad news. Table 5 provides insight into how social network analysis and related techniques are significant for crisis management.

### 4.3.2. Summary

Similarly, the analysis of existing studies shows the effectiveness of SNA tools and techniques in crisis communication and management. Specifically, different solutions grounded in SNA have helped address various solutions impacting overall crisis management. For example, SNA helps analyze social media data for knowledge purposes. SNA improves collaboration and crisis information seeking and sharing between stakeholders. Furthermore, SNA detects unfolding disasters such as earthquakes to improve an effective situational awareness system. In addition, the application of SNA is used to identify damage-related posts, identify types of users (influencers or followers), map user's activity, and observe ethnographics of users, and it is used for visualization purposes. Hence, addressing these issues through SNA tools and techniques has collectively helped in planning, communicating, managing, and coordinating effective and sustainable crisis management and decision-making processes.

## 5. Discussion and Open Issues

There are overlapping crisis management concepts, especially in developed businesses, which do not fully address the stakeholder perspective and sustainability [114]. The main focus of this study is to evaluate how ML, SNA, and related solutions are reshaping sustainable crisis management and decision making based on the existing literature on crisis informatics in order to address the research objective sufficiently for the researchers with the potential to make significant contributions to the crisis informatics literature. The study was designed based on three research questions, and the findings were discussed in the preceding section. Moreover, the literature analysis provides knowledge concerning the challenges and issues that remain unanswered with the potential to attract future studies. These themes are discussed according to the following;

### 5.1. Big Data and Language Support

In addition to the numerous other responsibilities that crisis management services take on, controlling outreach information and communication is becoming increasingly vital [115]. Crisis response generates information that contributes to big data, which supports crisis management efforts in providing an effective response and situational awareness [116]. Crisis management systems require data acquisition, processing, storage, and data use for decision purposes [117]. The data can be acquired from social media, disaster response platforms, and telecommunications networks. Because of awareness and decision-making activities, the precrisis situation is critical for the crisis response team [118]. According to the work of [119], one widely accepted piece of wisdom in crisis communication and management is that the best way to manage a crisis is to avoid one from occurring in the first place. If a crisis is avoided, neither the organizations nor the stakeholders suffer any consequences. The "alpha" or starting point of crisis management and crisis communication in crisis prevention [119]. Situational awareness can prevent or reduce the impact of crises [120]. The data or information generated from a crisis response can be analyzed to avoid or mitigate future occurrences. The perspective of the humanitarian community on the reliability of information collected from social media also raises concerns.

Big data can inform preparation and precrisis efforts positively. However, the contribution of big data to recovery efforts is lacking, demonstrating the need for further

research [116]. Identifying big data in crisis management comes with opportunities, yet the inherent challenges from this concept are still emerging in the crisis informatics literature. These challenges include capturing data, data storage, data analysis, and data visualization [121]. In addition, information processing operation challenges can be mapped into classifying, filtering, ranking, extracting, aggregating, and summarizing [38,50]. According to [38], identifying actionable information is significant and complicated, as the data is from many sources and in various formats. Universal cooperation is encouraged through logistics theory [95]. Similarly, since most datasets are in English, data limited to English could require revision to extend to other languages.

*5.2. Cross-Platform Support*

Among the challenging aspects of ML approaches in crisis informatics are the solutions to support multiple social media platforms. The elements that influence the acquisition and sharing of health-related information and the scope to which they apply are highlighted in [122]. Additionally, a study conducted by [123] examines WeChat users' social crisis information-sharing behavior and elucidates their decision-making process. Another study, using the SVM algorithm, demonstrates how a social position affects the collective sense-making process in crisis communication [28]. Additionally, a four-phase approach was created for analyzing crisis-management material curricula for teaching [124]. Numerous studies have demonstrated that academic and industrial research is insufficient to handle crisis informatics' concerns and challenges. Furthermore, multiple investigations are limited by the possibility of selection bias on social networking sites.

Moreover, this study identified Twitter as the most prevalent data source. Twitter is an interactive social media network founded in 2006 that enables users to communicate with 140 characters [125]. Researchers in crisis management have utilized Twitter for research purposes. This study identified 24 papers that used Twitter to conduct crisis management and communication research (see Figure 5). The study discovered that most publications concentrated on tweets' content, while others created tools that mine tweets for SNA. The discipline of Twitter-based crisis communication research is expanding; however, additional work are required to assist in developing standardized reporting criteria for Twitter research to increase its repeatability and comparability.

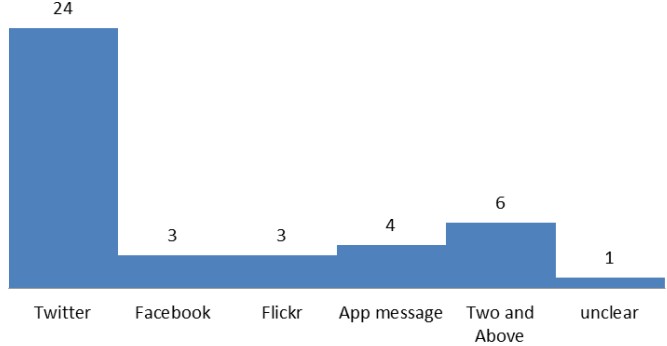

**Figure 5.** Occurrence of social media platforms

Nevertheless, this study substantiated [126]'s statement that the literature exaggerates the importance of Twitter. However, Facebook is the most popular social network globally, with 2.271 billion users, while Twitter has 326 million users and ranks 12th [127]. According to [126], Twitter has received nearly 500 times the attention it deserves. Moreover, Ref. [39] concluded in their study, entitled, 'comparative study of Facebook and Twitter', that Facebook is a more effective crisis management tool than Twitter. More research is needed from academia and industry focusing on broader social media platforms and empirical investigations [39]. The frequency of existing studies conducted with data from social networking platforms is presented in Figure 5.

*5.3. Dataset Access*

The inaccessibility of the dataset of previous work [103] is a big challenge inherent in the crisis informatics literature. An important criterion in assessing a research article is to check if the report is based on the work of others [128]. Any research that focuses on robust indicators of validity has a higher chance of being replicated" [129]. The engagement of scholars with prior work benefits from the logic, ideas, and findings established previously, thus avoiding the need to 'reinvent the wheel' [130]. By knowing prior work, Ref. [130] added that it would avoid the paths our predecessors showed to be dead ends. The absence of the dataset of crisis-related research is a significant challenge facing the area. According to [130], new theories or new empirical facts may push researchers into prolific fields of inquiry and break relationships considerably with previous work. Researchers are wary of a paper that does not build directly on prior work. This is because any new idea or finding must be adequate to attract current theories and logical rationale from their current interest. Hence, social media is not without any challenges as there is more significant challenge in using social media due to complexity, privacy, and third-party consent. For example, collecting disaster-related data for analysis is complex, and a standard crisis dataset is lacking. The provision of these will help build a better sentiment model and help in the accurate evaluation of the public's needs during and after a crisis [29].

## 6. Critical Reflection and Implication

*6.1. Crisis Management*

The analysis and synthesis of existing studies has identified various knowledge concerning the crisis informatics domain and the significance of various technological tools and techniques used to advance sustainable crisis management and communication. The literature synthesis has focused on crisis informatics research embracing ML, SNA, or related technological solutions. Specifically, studies focusing on improving decision making are mostly reported. This is obvious since decision making is vital for any organization to succeed in any industry. Likewise, decision making is critical for sustainable crisis management, from the initial precrisis phase until the recovery stage. In addition to decision support, sustainable crisis management involves information management (storage, processing, sharing, and dissemination), communication, response, people, and other tangible or intangible divisions of crisis or disaster. More specifically, some studies focus on citizen cooperation detection, location-based services, coordination and collaboration, intercommunal emotions and expressions, and resilience. Moreover, improving situational awareness through early warning and planning is the central focus of a few existing studies.

Moreover, emergent themes are significantly becoming the subject of a few studies. These themes include the social position to influence collective sense making, data collection via crowdsensing, damage assessment via crowdsourcing, critical infrastructure, and marketing and branding. Figure 6 depicts the summary of the various areas addressed by the current literature which still need improvement and are attracting future studies.

*6.2. Decision Making*

Numerous studies have demonstrated the efficacy of applying ML techniques for crisis management. For instance, Refs. [29,30] mined social media data for decision-making reasons. The [29] first strategy (ML and sentiment analysis) categorizes and classifies data to facilitate decision making, particularly during the crisis and recovery phases (postcrisis). The hybrid method described mining crisis-related data to find and identify vulnerable individuals. Additionally, Ref. [93] developed a methodology for collecting, storing, and analyzing data for decision making, while [116] investigated the influence of big data from social media. Furthermore, Ref. [28] examines how the public communicates uncertainty on social media during times of crisis and the effect of social positions on collective sense making. Numerous options, however, remain unavailable to meet the holistic picture and dynamic nature of crisis data to improve the sustainability of crisis communication.

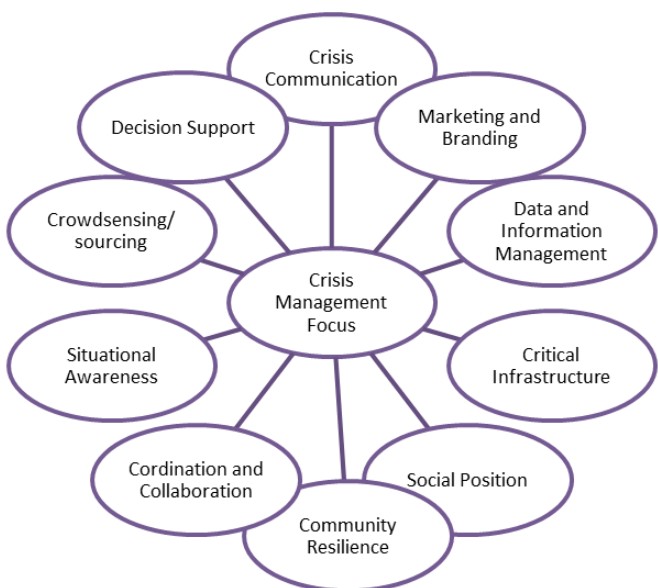

**Figure 6.** Implication of ML and SNA solutions.

### 6.3. Information Management

The work by [131] suggested a need to focus on the various ethical, legal, and social issues (ELS) relating to existing and emerging technologies for crisis response and management, since IT-based solutions are essential. This will enhance understanding of technology-based crisis management solutions, the collaborative mobilization of stakeholders and experts for decision support, avoid misinformation sharing and communication, improve social awareness about the situation, and enhance collaborative systems design and development for sustainable crisis management. Future studies may focus on domain adaptation and transfer, online and active learning, deep learning applications, situational awareness to actionable insights, and humanitarian disasters and health [50]. In addition, researchers should prioritize developing an ontology tailored to the public's needs and developing a lexicon of disaster-related keywords [29]. Moreover, it is also necessary to collect, store, and manage messages and tweets from various crises or disasters for data classification and provide a dataset of efficient standards for evaluation. This will significantly enhance the sustainability of information management. Thus, it is important to manage diverse outreach information and communication for sustainable recovery.

### 7. Conclusions

Crisis informatics encourages stakeholders to generate and share information of diverse categories in the name of crisis response. The information or data are harvested to advance the effort of sustainable crisis management through various technology tools and techniques that are still new and emerging due to the dynamic features of stakeholders' responses and the crises. The focus of this study is to review the ML techniques and related technologies used for emergency response in ways that prevent, reduce or mitigate a crisis. The study address three research questions: The study identified several areas of crisis informatics research, such as crowdsourcing, social media, AI and ML, news and blogs, and mashup, which answered the first question. AI and ML techniques cover supervised learning tools, unsupervised learning tools, deep learning, computer vision, and computation logistics, which addressed the second research question. The network analysis tools or related techniques comprise software, semantic analysis, textual analysis, thematic code, code frame, constructive rule, and social media API techniques, which answered the third research question. Hence, the study shows that modern analytics technologies enable data-driven crisis informatics and ML tools to reshape sustainable crisis response management and decision making. Finally, the study discussed various unanswered challenges, including big data and language support, cross-platform support, and dataset insufficiency.

*Limitation and Future Work*

The study's limitation is the selection bias associated with articles published in journals with a high IF impact factor. We utilized the IF ranking as a criterion for selecting and evaluating the articles' quality. The second constraint is exclusion criteria for conference proceedings, workshops, and book chapters. We think that the papers in such publications are primarily repetitions of ideas, concepts, or works-in-progress to be published as journal articles sooner or later. Moreover, disaster response was not used as one of the search keywords. Future studies may include this keyword to capture more studies. In addition, the focus of this study is to review how advanced technologies are used in respect to emergency response in ways that prevent, reduce or mitigate a crisis. Thus, future studies could classify the contributions that helped prevent, reduce or mitigate the crisis or disasters. Additionally, this study may be limited because journal databases such as MDPI and Emerald were not considered during the initial article search, although these databases are popular, reputable, and publish high-quality research articles. More importantly, these databases publish issues that are more relevant to crisis management and communication [132–136] than databases such as IEEEXplore. Therefore, future studies may target these journal databases for more relevant articles about crisis management and communication.

**Author Contributions:** This study was designed, directed, and coordinated by M.A.J., F.S. and M.A.J., as the principal investigator, provided conceptual and technical guidance for all the aspects of the project. F.S. assisted with financial support. U.A.B. planned the SLR and analyzed the data with R.N.H.N. and S.A.; U.A.B. and R.N.H.N. generated and synthesized the literature, and S.A. assisted in the supply and acquisition of the literature. I.I. and A.A. suggested and commented on the design of the study. The manuscript draft was written by U.A.B. and M.A. and commented on by all authors. All authors have read and agreed to the published version of the manuscript.

**Funding:** This work was supported by the Ministry of Higher Education through the Fundamental Research Grant Scheme under Grant FRGS/1/2020/ICT06/UPM/02/1.

**Acknowledgments:** The authors acknowledge Universiti Putra Malaysia and the Faculty of Computer Science and Information Technology, Malaysia, for all the support.

**Conflicts of Interest:** The authors declare no conflict of interest.

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
