# Peer review of "How Advanced Technological Approaches Are Reshaping Sustainable Social Media Crisis Management and Communication: A Systematic Review"

_sustainability, doi:10.3390/su14105854_

Round 1

Reviewer 1 Report

The article is an ambitious attempt at a review of how advanced data analysis approaches such as machine learning and social network analysis can assist crisis management communication. However, it has some shortcomings that need to be addressed.

The introductory section and literature review do not provide a satisfactory motivation for focusing only on machine learning and social network analysis – there is no mention of alternative approaches that can be applied to crisis management. Moreover, it seems that the authors do not distinguish between them ML and SNA, information communication technologies and social media. Thus, ML and SNA should be better explained and contextualised. A better explanation is needed also for other technological concepts that appear in the text (e.g., big data) instead of using them as buzzwords. Since the content of the paper is mostly about social media, the authors should consider changing the title so that it emphasizes that instead of ML and SNA.

In addition, the authors should explain how the paper is related to the topic of the special issue (i.e., the role of big data in sustaining open innovation strategies). There is no mention of open innovation in the paper and the use of the word sustainable seems forced in the title - apart from the introduction and conclusion this concept does not appear in the text.

Another issue is that two references are missing in section (question marks in square brackets).

The methodology section provides a good explanation of the methods used to identify research papers, select and quality assess. However, it is not clear why the search keywords did not include the keywords »machine learning«, »social network analysis and »sustainability«. The explanation of the data extraction and synthesis procedure (Section 3.4) is not comprehensive enough – what do the green and red colour in Figure 3 represent? Moreover, this is not consistent with Figure 1 that indicates 207 papers were extracted for full text reading. In addition, Figure 2 could be improved by providing exact frequencies for different categories.

The results are very descriptive and not well presented. The taxonomy in Figure 4 does not make a lot of sense. The content should be better explained and contextualised. In general, the paper is not well-structured. A lot of the content in the results section would need to be presented in the literature review section.

In the discussion, the authors do not provide a good explanation for the prevalence of Twitter as a data source. Maybe they should consider the data availability issue – it is not easy to scrape data from other platforms. Figure 5 is not well designed and is missing a scale and frequencies. Next, Figure 6 does not have any added value. It only lists different topics, without a critical reflection on relations between them.

Regarding language and style my impression is that the paper was not properly proofread. I recommend an extensive editing.

Finally, at the end of the paper there is no explanation of author contributions and no data availability statement.

Author Response

Dear Respected Reviewer,

We thank you for reviewing our manuscript and providing comments and pointing out suggestions to improve the paper. Based on the suggestions, we revised the manuscript. Our point-by-point response to comment is attached.

Reviewer 2 Report

The authors used crisis and disaster interchangeably in some places. Why do the keywords not include 'disaster response', which is commonly used by the public more frequently than the word 'crisis response'?

"The focus of this study is to review the ML techniques and related technologies used in respect to emergency response in ways that can prevent, reduce or mitigate a crisis."- In this case, the authors could have classified the articles which helped to mitigate/prevent and reduce the crisis or disasters.

"Moreover, it is also necessary to collect, store, and manage messages and tweets from various crises or disasters for data classification and provide a dataset of efficient standards for evaluation. This will significantly enhance the sustainability of information management." – Did any articles discuss the methods or systems to build sustainable information management.

"The research is funded by the Ministry of Education of Malaysia under the Fundamental Research Grant Scheme (FRGS)  FRGS/1/2020/ICT06/UPM/02/1". Whereas the authors have mentioned there is no external funding. - Kindly confirm.

Minor grammar corrections

Line No 536 ", which answered the first questions."

Line No 525 "recovery sustainability".- Sustainable recovery may be considered.

Author Response

(The authors gave the same response as above.)

Reviewer 3 Report

The paper covers just a literature review, this topic is really interesting, but authors are strongly requested to present precisely answer the question how for example big data support crisis management.

Taking into account the content of this paper I say I learn nothing. The considerations are very general, too general to be useful. Therefore, authors are requested to add experiments. It is necessary to make this study useful and interesting to readers

Author Response

(The authors gave the same response as above.)

Reviewer 4 Report

The article is more of a review article than a research article. An overview of the methods and models used to solve various problems of economic practice is given.

But, firstly, the article lacks a generalization and understanding of these approaches from the standpoint of their strengths and weaknesses in the context of solving specific substantive problems.

Secondly, the type of research problem the work is aimed at is not given.

Thirdly, the article contains only a list of approaches and tools for decision-making in crisis situations. But there is no critical reflection and recommendations for their use.

In addition, there is no empirical study, no interpretation of it.

Thus, the work contains a list of existing cases and works of other authors without generalization and conclusions. The article does not contain the author's idea, the concept of using data analysis methods to solve applied problems.

Author Response

(The authors gave the same response as above.)

Round 2

Reviewer 3 Report

It's a pity that it is just Literature Review, however it is well done

Author Response

Dear Respected Reviewer,

We thank you for reviewing our manuscript and providing comments to improve the paper. Based on the suggestions, we revised the manuscript. Our response to comment is attached.

Reviewer 4 Report

Review round 2.

Comments and suggestions:

  1. I recommend the authors to add an overview of social network mining methods and tools
  2. The analysis presented in fig. 4 can be supplemented with approaches that are currently not common for the analysis of social networks, but could be useful for this
  3. The analysis presented by the authors in tables 1-4 does not contain generalizing provisions. It is necessary to introduce generalizing provisions, this will improve the quality of the material presented.

Author Response

(The authors gave the same response as above.)

Round 3

Reviewer 4 Report

All comments of reviewer have been taken into account by the autors

Author Response

.